# Enzyme Inhibition-Based Assay to Estimate the Contribution of Formulants to the Effect of Commercial Pesticide Formulations

**DOI:** 10.3390/ijms24032268

**Published:** 2023-01-23

**Authors:** Elena N. Esimbekova, Valeriya P. Kalyabina, Kseniya V. Kopylova, Victoria I. Lonshakova-Mukina, Anna A. Antashkevich, Irina G. Torgashina, Kirill A. Lukyanenko, Elena V. Nemtseva, Valentina A. Kratasyuk

**Affiliations:** 1Institute of Fundamental Biology and Biotechnology, Siberian Federal University, 660041 Krasnoyarsk, Russia; 2Laboratory of Photobiology, Institute of Biophysics of Siberian Branch of Russian Academy of Science, 660036 Krasnoyarsk, Russia; 3Laboratory of Cell Molecular Physiology and Pathology, Scientific Research Institute of Medical Problems of the North of the Siberian Branch of the Russian Academy of Sciences, 660022 Krasnoyarsk, Russia; 4Laboratory for Digital Controlled Drugs and Theranostics, Federal Research Center “Krasnoyarsk Research Center” of Siberian Branch of Russian Academy of Science, 660036 Krasnoyarsk, Russia; 5Laboratory for Biomolecular and Medical Technologies, Krasnoyarsk State Medical University after Prof. V.F. Voino-Yasenetsky, 660022 Krasnoyarsk, Russia

**Keywords:** enzyme inhibition-based assay, pesticides, bioluminescent assay, luminous bacteria, conjugated enzyme reactions, formulants

## Abstract

Pesticides can affect the health of individual organisms and the function of the entire ecosystem. Therefore, thorough assessment of the risks associated with the use of pesticides is a high-priority task. An enzyme inhibition-based assay is used in this study as a convenient and quick tool to study the effects of pesticides at the molecular level. The contribution of formulants to toxicological properties of the pesticide formulations has been studied by analyzing effects of 7 active ingredients of pesticides (AIas) and 10 commercial formulations based on them (AIfs) on the function of a wide range of enzyme assay systems differing in complexity (single-, coupled, and three-enzyme assay systems). Results have been compared with the effects of AIas and AIfs on bioluminescence of the luminous bacterium *Photobacterium phosphoreum*. Mostly, AIfs produce a considerably stronger inhibitory effect on the activity of enzyme assay systems and bioluminescence of the luminous bacterium than AIas, which confirms the contribution of formulants to toxicological properties of the pesticide formulation. Results of the current study demonstrate that “inert” ingredients are not ecotoxicologically safe and can considerably augment the inhibitory effect of pesticide formulations; therefore, their use should be controlled more strictly. Circular dichroism and fluorescence spectra of the enzymes used for assays do not show any changes in the protein structure in the presence of commercial pesticide formulations during the assay procedure. This finding suggests that pesticides produce the inhibitory effect on enzymes through other mechanisms.

## 1. Introduction

The increasingly extensive application of agrochemicals for protecting crops and enhancing their productivity is an incentive to the development of new approaches and techniques in pest control. With the growing tendency for sustainable agriculture, obsolete and dangerous pesticide formulations are replaced by safer and more effective ones [1], alternative approaches are suggested to using the existing chemicals [2], and short-half-life and slow-release formulations are developed [3]. Controlled pesticide release systems are a promising strategy for substantially reducing pesticide losses in the environment owing to targeted delivery of the pesticides to pests [4,5]. These approaches can effectively reduce the pressure of side effects of pesticides on the environment and non-target organisms. 

However, the safety of formulants—substances added to pesticide formulations (co-solvents, adjuvants, safeners, etc.)—remains an issue and currently attracts considerable research attention. No measures are taken to mitigate the effects of formulants on non-target organisms, there are no restrictions on the use of formulants [6], and they are not regularly tested for environmental toxicity [7].

At the same time, various assays performed in a number of studies convincingly demonstrated the difference between the effects of active ingredients alone (AIas) and as components of pesticide formulations (AIfs). Numerous studies show that herbicide [8], fungicide [9], and insecticide [10] formulations can be more toxic than their active ingredients. For example, solvents and added adjuvants considerably contributed to the toxic effect of the Focus^®^ Ultra herbicide formulation (with cycloxydim as the active ingredient) on amphibians [11]. The data reported in a study by Straw and Brown [12] suggest that toxicity of the Amistar^®^ fungicide (with azoxystrobin as the active ingredient) for bees is mainly caused by alcohol ethoxylates. The extensive systematic review by Nagy et al. [13] shows that more than 50% of the studies analyzed by the authors reported higher toxicity levels of pesticide formulations compared to their active ingredients. Formulants are potentially capable of modifying their own toxic properties and reactivity as they move along trophic pathways, thus becoming even more dangerous for non-target organisms. Some data suggest that surfactants and adjuvants contribute to an increase in cytotoxicity of pesticide formulations by enhancing bioavailability of the active ingredients [14,15]. In addition, safeners—components added to protect crops from the toxic effect of herbicides—are capable of reduction transformations into herbicide-like products [16], or they can produce their own toxic effects, e.g., by changing the activities of oxidative stress enzymes in aquatic organisms [17]. An indirect contribution of formulants to toxicity was shown for crustacean *D. magna* [18], luminous bacterium *P. leiognathi*, green microalga *P. subcapitata* [19], common frog *R. temporaria* [11], annelid worm *E. albidus* [15], and gastropod mollusk *H. tuberculate* [20].

According to the adverse outcome pathway (AOP) concept [21], clinical and subclinical signs of toxic injury are consequences of the toxic effects at the molecular level. Enzymes are among the most significant and research-worthy endpoints of the effects of pesticide formulations [22]. Enzymatic reactions responsible for various metabolic processes (such as conduction of nerve impulse, metabolism of biomolecules, and trace element metabolism) were found to be sensitive to pesticides and, hence, effective for assessing potential harmfulness of pesticides for organisms. Moreover, such sensitivity to pesticides provided the basis for developing various enzyme-based biosensors [23,24]. 

This work is the continuation of our previous study, which was conducted to compare the effects of commercial pesticide formulations on the in vitro and in vivo functions of assay systems [25]. The purpose of the present study was to reveal the contribution of formulants to the toxicity of commercial pesticide formulations. An enzyme inhibition-based assay was used to compare the effects of active ingredients of pesticides and commercial formulations based on those active ingredients on enzyme systems. Tests were performed with enzyme assay systems of different complexity: (1) single-enzyme reactions catalyzed by alkaline phosphatase (ALP), NAD(P)H:FMN-oxidoreductase (Red), alcohol dehydrogenase (ADH), butyrylcholinesterase (BChE), lactate dehydrogenase (LDH), and trypsin; (2) multi-enzyme reactions catalyzed by the coupled system of luminous bacteria NAD(P)H:FMN-oxidoreductase + luciferase (Red + Luc) and the three-enzyme systems of lactate dehydrogenase + NAD(P)H:FMN-oxidoreductase + luciferase (LDH + Red + Luc) and alcohol dehydrogenase + NAD(P)H:FMN-oxidoreductase + luciferase (ADH + Red + Luc). Different chemical classes of pesticides (organophosphorus, pyrethroid, and neonicotinoid compounds) were tested in the current study. We determined the enzymes that were the most susceptible to the effects of various classes of pesticides and found an increase in the inhibitory effect of AIfs relative to AIas. The results were compared to the effects of AIas and AIfs on bioluminescence of luminous bacterium *Photobacterium phosphoreum* (*P. phosphoreum*). The effects of pesticide formulations on the structure of the enzymes were tested using optical spectroscopy methods.

## 2. Results

### 2.1. Physico-Chemical Characterization of Active Ingredients of Pesticides

As many active ingredients of pesticides are water insoluble, we prepared their solutions using not only distilled water but also ethanol and acetonitrile. Our previous studies [25,26] showed that these organic compounds effectively dissolved active ingredients of a number of pesticides (OPs, pyrethroids, neonicotinoids) and commercial formulations based on them, exerting the minimal inhibitory effect on the enzyme-based assay systems. In experiments with the assay system based on the luminous bacterium, solvents were distilled water and ethanol, as their inhibitory effect on bacterial luminescence was less pronounced than that of acetonitrile. 

The enzyme systems exposed to the effects of toxicants in the present study have certain properties affecting optical signal measurements. The absorption properties of the analyzed samples can distort the measured activity of the enzyme systems. Therefore, we studied absorption spectra of various concentrations of the active ingredients and pesticide formulations based on them.

AIa solutions exhibited absorption mainly in the UV range (of the wavelength < 300 nm) and had low extinction (Figure 1, solid lines). Thus, AIa absorption could hardly distort the measurement of the enzyme activity, except for trypsin. AIf absorption was found to be significantly higher compared to AIa of the same concentration because of the additional components (formulants) of pesticide formulations (Figure 1, dashed lines). However, again, that effect was more pronounced for the wavelength of measuring trypsin activity only.

In subsequent experiments, we varied the AIa and AIf concentrations within the range where optical density of the sample at the measurement wavelength did not exceed 1.

### 2.2. A Study of Sensitivity of the Assay Systems to the Active Ingredients of Pesticides

The principle of the enzyme inhibition-based assay, which underpins this study, is detection of changes in enzyme activity in the presence of potentially toxic compounds compared to the reference values. The results imply a conclusion about the functional changes in reaction components caused by exposure to the tested substances. In the present study, we estimated the effects of high-purity active ingredients of pesticides on the function of single-enzyme and multi-enzyme reactions and luminescence of the *P. phosphoreum* bacterium. We obtained concentration dependencies of enzyme activities in the presence of seven AIas representing three groups (organophosphorus, pyrethroid, and neonicotinoid compounds) and determined IC_50_ values (Table 1).

The addition of some of the AIas to the reaction mixture resulted in a decrease in enzyme activity (Table 1). The most pronounced inhibitory effect on the function of both single- and multi-enzyme assay systems was produced by pyrethroids (cypermethrin, deltamethrin, and fenvalerate). For instance, the IC_50_ of cypermethrin acting on the single- and three-enzyme assay systems based on ADH was 0.2 mg/L, which corresponded to maximum residual levels (MRLs) of this compound in fruit and vegetables. At the same time, exposure to pyrethroids slightly stimulated the activities of the enzyme assay systems based on two hydrolases—trypsin and ALP. For example, the presence of 10 and 13.5 mg/L of fenvalerate resulted in a 28% and 26% increase in trypsin and ALP activities, respectively.

The two other AIa classes produced a considerably weaker inhibitory effect on the assay systems: concentrations of OPs (diazinon, glyphosate) that caused a 50% decrease in enzymatic activity were several orders of magnitude higher than the MRLs of these compounds in fruit and vegetables. We failed to estimate the effect of malathion on most enzyme assay systems: when malathion was added to the reaction mixture, the solution became colored and turbid, which prevented accurate analysis. Neonicotinoid imidacloprid added in concentrations within the tested range (0.01–1 g/L) did not inhibit the activities of the enzyme assay systems (Table 1). None of the seven high-purity AIas added in concentrations within the tested range produced any effect on the activity of Red and bioluminescence of *P. phosphoreum*. No inhibitory effects were produced on LDH by diazinon, on ADH by fenvalerate, or on trypsin by cypermethrin. It was impossible to increase concentrations of those AIas substantially because of their poor solubility in the solvents used and a greater interference of the optical effects of AIa solutions in results of the assay. Moreover, there was no point in assessing the effects of larger concentrations of the tested AIas on the assay systems because they would be considerably higher than their MRLs in foods.

There was certain specificity in enzyme responses to AIas with different targets. For example, sensitivity of single-enzyme reactions to the glyphosate herbicide decreased as follows: BChE > trypsin > ALP > ADH > LDH/ Red (Figure 2); moreover, no 50% inhibition of the last two reductases was achieved in the tested concentration range. At the same time, LDH, which was weakly affected by OPs, had a noticeably lower activity in the presence of any of the pyrethroids. Among the single-enzyme reactions, the assay system catalyzed by ADH was the most sensitive to exposures to AIas.

As the chain of conjugated enzyme reactions was elongated (the coupled enzyme system Red + Luc and the three-enzyme systems ADH + Red + Luc and LDH + Red + Luc), the inhibitory effects of certain AIas were stronger. For instance, sensitivity of assay systems to deltamethrin increased as follows: ADH ˂ Red + Luc ˂ ADH + Red + Luc; the IC_50_ values were 10.4, 3.7, and 1.0 mg/L, respectively (Figure 3).

### 2.3. Comparing Effects of Commercial Pesticide Formulations and Their Active Ingredients on the Function of Assay Systems

Next, we compared the effects of AIas and AIfs on the function of the assay systems. The results obtained indicated a substantial difference between the effects of AIas and AIfs on assay systems.

In some experiments, pyrethroids such as deltamethrin and cypermethrin as AIas had stronger inhibitory effects on assay systems than the pesticide formulations containing these active ingredients. For instance, a 50% decrease in the enzymatic activity of ADH was observed in the presence of 0.2 mg/L of cypermethrin, which was lower by a factor of 500 than the value obtained for the Briz commercial formulation based on cypermethrin (Table 1). Similar results were obtained in experiments with multi-enzyme systems exposed to deltamethrin and cypermethrin. The average IC_50_ values of these AIas were one order of magnitude lower compared to the corresponding values of the AIfs. For example, for the Red + Luc coupled enzyme system, the IC_50_ values of deltamethrin as the active ingredient of the Delcid pesticide and deltamethrin alone were 39.5 and 3.7 mg/L, respectively (Table 1). 

However, these results were the exception rather than the rule. In most experiments, the effects of commercial formulations on assay systems, multi-enzyme ones in particular, were considerably stronger. The difference in the effects of OPs was clearly demonstrated in experiments with the Red + Luc coupled enzyme system: a 50% inhibition of the assay system activity was observed in the presence of 0.3 g/L of glyphosate as AIa, and the concentration of glyphosate as the component of the Tornado Extra formulation needed to achieve the same effect was lower by a factor of 160 (Figure 4). For another OP compound, diazinon, the difference between the inhibitory effects of AIa and AIf on multi-enzyme assay systems was even more pronounced: the IC_50_ of diazinon as the active ingredient of the Muravyed formulation for LDH + Red + Luc was six orders of magnitude lower than the IC_50_ of diazinon alone. A similar result was obtained for a fenvalerate pyrethroid: the sensitivity of the Red + Luc assay system to AIf as the active ingredient of the Sempay formulation was three orders of magnitude higher than to fenvalerate as AIa (Figure 5).

The comparison of the effects of AIas and AIfs on assay systems suggested a considerable contribution of additional components to the inhibitory effect of the pesticide formulation. For example, the imidacloprid neonicotinoid used as AIa did not produce any detectable effect either on enzyme activity or on intensity of bioluminescence of the luminous bacterium regardless of its concentration within the tested range (Table 1). However, commercial pesticide formulations based on imidacloprid demonstrated noticeable inhibitory effects on assay systems: high sensitivity to imidacloprid MRLs in fruit and vegetables was observed in experiments with single-enzyme reactions based on ADH, Red, and LDH and all multi-enzyme reactions. 

The contribution of formulants to inhibition of enzymatic assay systems was additionally supported by the differences in IC_50_ that we previously found for commercial pesticide formulations with the same active ingredient [25]. For instance, formulations containing imidacloprid (Biotlin, Corado, and Confidor Extra) differed in their inhibitory effects on the same assay systems by a factor of 600 for single-enzyme systems and 11,000 for multi-enzyme ones. 

In a number of tests, there was a correlation between responses of assay systems to AIa and AIf. Trypsin showed sensitivity to both high-purity glyphosate and commercial formulations based on it. The sensitivity of the trypsin-based assay system to glyphosate increased as follows: Liquidator < Tornado Extra < glyphosate. 

The transition from single-enzyme to coupled and three-enzyme assay systems was interesting in that sensitivity to the inhibitory effects of toxicants was expected to increase, but it was not always the case. For instance, in the presence of the Sempay commercial formulation, the sensitivity of assay systems was enhanced as the enzyme conjugation chain was elongated (the IC_50_ values of fenvalerate as AIf were decreased as follows: LDH > Red + Luc > LDH + Red + Luc (Table 1)). By contrast, no such tendency was observed for fenvalerate as AIa (IC_50_ values were decreased in reverse order, and the lowest value was obtained for the LDH-based single-enzyme system). Similar results were obtained for cypermethrin as AIa and as AIf in the Briz commercial formulation (Figure 6).

The effects of the active ingredients on bioluminescence of *P. phosphoreum* were either insignificant or undetectable within the range of the tested concentrations: no EC_50_ value was obtained for any of the AIas. Moreover, the effect of 20% quenching of bacterial bioluminescence was only observed for imidacloprid. The EC_20_ values for imidacloprid as AIa and as AIf in the Confidor Extra formulation were 50 and 30 mg/L, respectively. 

These findings suggest that high-purity active ingredients of pesticide formulations affect the function of enzyme-based assay systems, but the sensitivity of the assay systems to them differs substantially from the sensitivity of the assay systems to commercial pesticide formulations. Most tests demonstrated stronger effects of the commercial formulations on all assay systems.

### 2.4. The Effect of Pesticide Formulations on the Structure of Enzymes Used in the Assays

Since one of the possible mechanisms of the inhibitory effect of a toxicant on enzymatic reactions is disruption of the structure of enzymes, we examined changes in optical spectra of the proteins used in the assays caused by the presence of commercial pesticide formulations. Circular dichroism spectra were measured to reveal an alteration of the secondary structure of proteins, while fluorescence spectra were studied to find the change in their tertiary structure. The enzymes exhibiting high sensitivity to the commercial pesticide formulations, namely, ADH, LDH, BChE, ALP, and trypsin, were chosen for this study (Table 1). The Sempay, Muravyed, Delcid, Liquidator, Biotlin, and Tornado Extra commercial pesticide formulations were used as inhibitors. Each enzyme was studied in the corresponding buffer used before for the activity measurement. The pesticide formulations were preliminarily dissolved in water, buffer, or ethanol as described above. After addition of the formulation to the protein sample, the final concentration of ethanol did not exceed 5%. The spectra of protein sample with the added commercial pesticide formulation were compared to the spectra of the control sample with the addition of the appropriate amount of water, buffer, or ethanol. In each case, the ratio between the concentrations of enzyme and commercial pesticide formulations was close to that obtained for the 50% inhibition effect (Table 1). 

Circular dichroism spectra demonstrated that all proteins retained their secondary structure under conditions described above. The examples of CD spectra of ADH in the presence of the Muravyed commercial pesticide formulation or ethanol, as well as without any additives, are shown in Figure 7. All CD spectra of the ADH were found to have double minima at 208 and 220 nm, which is consistent with previously published data [28] and reflects the well-known α-helical structure of this protein.

The fluorescence spectra of the enzymes were measured under excitation of 280 and 290 nm. In the latter case, only tryptophan residues of the proteins are excited, as they are very sensitive to the polarity of the microenvironment, responding to the protein structure change or direct contact with co-solvents by alteration of fluorescence intensity and spectral distribution. Under excitation of 280 nm, tyrosine residues, which are sensitive to the local pH, fluoresce as well [29]. In the current study, we compared the spectral profiles and intensities of the protein fluorescence in the presence and in the absence of commercial pesticide formulations.

The comparison of the fluorescence spectra indicated that, for the majority of the enzymes used in this study, the pesticide formulations did not affect the protein tertiary structure and did not interact with the protein surface in the region of location of tryptophans. An example of LDH in the presence of the Muravyed formulation is shown in Figure 8a,b. The peak intensity of the protein fluorescence spectrum was observed at about 341 nm in the presence of both ethanol (control) and pesticide formulation. However, the fluorescence of BChE changed in the presence of both tested formulations—Tornado Extra and Biotlin. After addition of the Tornado Extra formulation, spectral maximum of BChE fluorescence was blue-shifted from 331 to 327 nm without intensity change as compared with control sample (Figure 8c,d). In the experiment with Biotlin, essential quenching of BChE fluorescence was observed (Figure 8e), with a slight blue shift of the spectral maximum to 329 nm (Figure 8f). That could indicate that the tryptophan residues of this enzyme had moved to a less polar environment in the presence of the pesticide formulations. Thus, the spectral study demonstrated that, under inhibition of BChE by Tornado Extra and Biotlin, direct interaction between enzyme surface and formulation components could contribute to the decrease in the reaction rate. 

For all proteins studied here, we observed the same profile of fluorescence spectra under 280 and 290 nm excitation, suggesting that tryptophan residues were the main fluorescence emitters of the proteins.

Thus, the fluorescence spectra and circular dichroism spectra obtained in the current study suggest that the effects of pesticide formulations on the tertiary and secondary structures of the majority of the proteins were insignificant. Thus, the inhibitory effect of commercial pesticide formulations demonstrated in the present study caused by different mechanisms such as the effect of pesticides on ligands (substrates of enzymes), disruption of the enzyme–substrate interaction (and, for multi-enzyme reactions, disruption of the interaction between enzymes in the conjugation chain) or some others.

## 3. Discussion

Potential environmental hazards associated with the use of pesticides on the global scale is the reason for increased research effort in this area. More and more data have been gathered showing that toxicity of pesticide formulations is underestimated. The cumulative effect of various factors leads to considerable losses of pesticides in the environment, making it necessary to apply greater amounts of pesticides [4]. As pesticide components differ in chemical nature, they have dissimilar environmental fate [30]. Pesticides interact with both biotic and abiotic environmental factors. For example, UV radiation may change toxicity of pesticide formulations [31]. One of the most important factors, however, is that pesticide formulations are composed of various compounds. 

The active ingredients of pesticides have been specially developed to disturb the function of essential processes in target organisms (enzymatic reactions, mitochondrial respiratory chain, macromolecule biosynthesis, etc.) [31]. However, because of complexity and diversity of molecular processes, the side effects of pesticide formulations on non-target species may damage their enzyme systems, physiological mechanisms, or the balance of the major biomolecules. Moreover, the damage may be unclear at the morphological and physiological levels, but it leads to long-term sublethal metabolic and genetic injuries [32]. There are data relating the effects of realistic concentrations of pesticides on bioassays to harmful effects at the level of biomolecules: lipid metabolic disorder [33] and changes in the levels of vital bioactive compounds [34] and rates of metabolic processes [35].

To reduce potential risks, pesticide application is regulated by statutory instruments. Risk assessment is mainly focused on active ingredients, which are responsible for pesticide toxicity. Recently, however, special consideration has been given to formulants—inert ingredients of commercial pesticide formulations that increase the effects of pesticides. The comparative studies of the effects of active ingredients and commercial formulations based on them conducted for glyphosate [8,36], diazinon [19], clothianidin [18], and other pesticides show quite convincingly that the stronger effects of pesticide formulations may be caused by the contribution of additional ingredients at the level of biological reactions. There are data, though, showing that the active ingredient alone was more toxic than the formulation, which could be attributed to the antagonistic effect [18].

As we discussed elsewhere [25,30], there are various reasons why the danger of formulants in pesticide formulations, where their percentage is incomparably greater than that of the active ingredients, is underestimated. Briefly, formulants, in contrast to active ingredients, do not usually undergo ecotoxicological tests, many manufacturers do not disclose their composition and concentrations, and their side effects remain insufficiently understood. No regulations define maximum admissible concentrations of these inert compounds in the environment. Quick degradation, which is an essential parameter of active ingredients for manufacturers of pesticides, is not a requirement for formulants, and, thus, the half-life of formulants in pesticide formulations is often considerably longer than the half-life of the active ingredients [37]. 

Although formulants are conditionally inert compounds, they are capable of contributing to the toxic effect, and their contribution is sometimes equal to or greater than the effect of the active ingredient. Not only can additional components increase the inhibitory effect of active ingredients of pesticide formulations, but they also can function as inhibitors of enzymatic activity. A number of researchers attributed stronger effects of pesticide formulations on non-target organisms to their effect on enzymes and cytotoxicity [38,39]. The wide diversity of formulants, which are added to improve certain parameters of pesticides (solubility, storage stability, penetration to tissues of pests, etc.), makes their potential toxicity even less predictable. For instance, mechanisms of toxicity of adjuvants at the molecular level seem to be related to destabilization of biological membranes [20].

The data obtained by our research team suggest that sometimes environmentally significant concentrations of high-purity active ingredients can potentially produce effects at the molecular level. For instance, the concentration of the cypermethrin pyrethroid that caused a 50% decrease in the enzymatic activity of ADH was close to its MRL in fruit and vegetables. Other enzyme assay systems were also affected, although less significantly, by the active ingredients of pyrethroids. 

Our previous study [25] showed that a number of commercial pesticide formulations substantially inhibited activities of different enzymes. Comparison of those data with the effects of high-purity active ingredients on assay systems provided an indirect estimate of the contribution made by formulants to an increase in the inhibitory effect of pesticide formulations. A study of fenvalerate activity clearly demonstrated differences between the effects of AIa and AIf. Enzymatic reactions were inhibited in the presence of fenvalerate as AIa, but the inhibitory effect was less pronounced than in the presence of the fenvalerate-based commercial formulation Sempay. The ADH + Red + Luc multi-enzyme system exhibited the highest sensitivity to fenvalerate as AIa: the IC_50_ value was 1.6 mg/L, which was, however, higher than the MRL of fenvalerate in food products. When fenvalerate was used as the active ingredient of the commercial formulation Sempay (AIf), the IC_50_ value was 0.0006 mg/L, i.e., three orders of magnitude lower than the admissible levels of fenvalerate in food products. 

Our results are consistent with the literature data. Changes in enzyme activities were observed after exposures to both active ingredients of pesticides (inhibition of carbonic anhydrase [40], an increase in activities of antioxidant enzymes [15]) and commercial pesticide formulations (inhibition of esterase, glutathione-s-transferase, glutathione reductase) [41,42].

The present study showed that the differences in the effects of formulations containing the same active ingredients but produced by different manufacturers that we noted previously [25] could be attributed to the dissimilar composition of formulants in those pesticide formulations. Of the seven active ingredients analyzed in this study, glyphosate had the strongest effect on BChE, with IC_50_ reaching 35 mg/L. When that hydrolase was exposed to glyphosate as AIf in the Tornado Extra formulation, the IC_50_ value was lower by a factor of 15, i.e., 2.4 mg/L. By contrast, the IC_50_ value for another glyphosate-based formulation, Liquidator, was greater by a factor of 29 compared to glyphosate as AIa, reaching 1 g/L. Hence, it is important to study the effects produced on assay systems by not only active ingredients of pesticide formulations but also formulants. These findings also highlight the complexity of molecular mechanisms underpinning clinical and subclinical effects of toxicants on living organisms. 

Quite often, the effects of pesticides on the activities of enzymes are not class specific. Different representatives of the same class of pesticides may have dissimilar effects on a certain enzyme. For example, acetylcholinesterase (AChE) activity was increased in response to the action of imidacloprid, but it was inhibited by other neonicotinoids such as guadipyr and cycloxaprid [22]. The reverse is also true: a commercial formulation can have different effects on different enzymes, and, in this context, enzymes have certain specificity. The commercial formulation Dursban (20% emulsified concentrate) based on chlorpyrifos decreased activities of ALP, AChE, and catalase (CAT) in fish liver, increasing activities of acid phosphatase (AP), aspartate aminotransferase (AST), and alanine transaminase (ALT) [43]. Previous results [25] and findings of this study suggest that the strongest inhibitory effect on dehydrogenases LDH and ADH was produced by pyrethroids, both as AIa and as AIf.

Effects of different pesticide classes were compared in experiments with assay systems growing in complexity—from simple single-enzyme reactions to more complex conjugated enzyme reactions and organisms. Our previous study demonstrated that as the length of the enzyme conjugation chain in assay systems was increased, the sensitivity of assay systems to commercial pesticide formulations increased by several orders of magnitude [25]. However, analysis of the effects of high-purity active ingredients of pesticides on assay systems showed that that trend was only observed for high-purity deltamethrin, whose inhibitory effect increased as follows: ADH < Red + Luc < ADH + Red + Luc. That was another substantiation of the essential contribution of formulants to the inhibitory effects of commercial pesticide formulations. The highest sensitivity to the pesticides tested in this study both as active ingredients alone and as components of commercial formulations was exhibited by the ADH + Red + Luc assay system. 

A possible mechanism through which various xenobiotics decrease the rate of enzymatic reaction may be molecular interaction between reaction components and the added compounds. Hydrogen bond formation and hydrophobic interactions with enzymes could cause a change in the conformation and charge of amino acid residues, which, in turn, alters the secondary and tertiary structures and results in a decrease in catalytic activity [44,45,46]. The general purpose of the methods used to establish mechanisms of enzyme inhibition by toxic substances is to solve two types of problems: to reveal formation of the bond between the enzyme and the inhibitor (formation of a stable complex) and to detect possible conformational changes in the structure of proteins in the presence of toxic substances. For these purposes, various optical techniques, including absorption, fluorescence, and circular dichroism spectroscopy, as well as molecular modeling methods are widely used [47,48,49]. 

We studied possible changes in the structure of the most sensitive enzymes (ADH, LDH, BChE, ALP, and trypsin) caused by interactions with components of the commercial pesticide formulation. Fluorescence spectroscopy was used to detect changes in the tertiary structure and circular dichroism spectroscopy in the secondary structure of the enzymes [50,51]. Trying to approach the conditions used for the enzyme inhibition-based assay, we maintained the ratio between the concentrations of enzyme and pesticide formulations close to that obtained for the 50% inhibition effect. In addition, measurements were taken immediately after mixing without incubation. Under those conditions, the signs of interaction between components of the commercial pesticide formulation and the protein were only observed in the experiment with BChE after addition of the glyphosate-based commercial formulations Tornado Extra and Biotlin. In the presence of pesticide solutions, the spectrum was shifted to the short-wavelength range. Biotlin also caused a decrease in fluorescence intensity by a factor of about 1.4. That could be indicative of the interaction between components of formulations and protein surface, where tryptophan residues were located. 

CD spectra of ADH, LDH, BChE, ALP, and trypsin were not altered in the presence of the pesticide formulations, although some studies using this technique revealed structural changes of proteins, e.g., of human serum albumin by fungicide carbendazim [52], of pepsin by pyrethroid insecticides [53], etc. CD spectra provide different information depending on the biological recognition element. For example, for aptamer-based sensors, circular dichroism spectroscopy is used to estimate the binding affinity of nucleic acid fragments against a certain pesticide [54], whereas, in enzyme inhibition-based assays, it is applied to detect a general change in the protein secondary structure, without any specificity [48,55].

The change of the intrinsic protein fluorescence in the presence of different xenobiotics is extensively used to study the action mechanisms of the toxic substances. The decrease in fluorescence intensity under variation of the temperature and additive concentration can be used as the basis for estimating the affinity and thermodynamic characteristics of protein–xenobiotics interaction (see [52,53,55] as examples). However, to the best of our knowledge, the direct interactions between enzymes and pesticides used in our work have never been studied.

Three reasons could be proposed to explain the absence of a denaturing effect of the studied pesticide formulations on the enzyme structure: (i) low concentration of the additives; (ii) a stabilizing effect of components other than pesticide components of the formulations; and (iii) too short time of incubation of the proteins with additives. Since this part of our study was aimed at elucidating the mechanism of the observed inhibitory action of the pesticide formulations, the experimental conditions were the same as those under which the activities of the studied enzymes were measured. A wider variation of the experimental conditions could result in pronounced disruption of protein structure by pesticide formulations, but this would be the subject of further detailed research. 

As the tertiary and secondary structures of most proteins were not disrupted, we assume that the main contribution to the inhibitory effect of pesticides on enzymes was made by other mechanisms such as interaction of pesticides with enzyme substrates and disruption of enzyme–substrate interaction or interaction between enzymes in the conjugation chain. 

The knowledge of the mechanisms of pesticide molecular action forms the basis for the methods of monitoring pesticide residues using biosensor processes, which, in addition to enzymes, employ such molecular recognition elements as antibodies, nucleic acids, aptamers, etc. [56].

The enzymes that are commonly used to detect pesticides include hydrolases AChE, BChE, alkaline phosphatase, lipase, as well as oxidoreductases horseradish peroxidase, tyrosinase, and laccase. Electrochemical biosensors based on AChE and horseradish peroxidase were effectively used to detect OPs: detection limits were 0.16 ng/mL of malathion and 0.025 mg/L of glyphosate, respectively [57,58]. In the current study, among the single-enzyme systems, the enzyme assay system with ADH exhibited the highest sensitivity to another OP pesticide—diazinon. The values of IC_50_ for diazinon as AIa and AIf were 14.5 and 0.2 mg/L, respectively. Enzyme biosensors based on multi-enzyme systems show considerable promise as well [59]: they exhibit high sensitivity to toxic substances, as confirmed by results of the present study. 

The principal advantages of immunosensors over the enzyme-based biosensors are the higher stability of antibodies/antigens used as recognition elements and greater selectivity and specificity. Modifications with different (nano)materials and the use of enzymatic tags make it possible to produce diverse immunosensors, which are capable of detecting pesticides in real food samples [60].

Aptamers (short nucleotide sequences of single-stranded ribonucleic or deoxyribonucleic acids) are used as the basis for developing specific, measurable, accurate, robust, and time-saving (SMART) biosensors—aptasensors. They demonstrate high selectivity in binding with targets and remain functionally active during long-term storage, even at room temperature [61]. Higher stability, longer lifetime, and lower cost are advantages of aptamers over enzymes and antibodies [60]. Aptasensors exhibited high sensitivity to pesticides such as fipronil [62], diazinon [63], chlorpyrifos [64], and acetamiprid [65] in real samples (fruits, vegetables, wastewater). The SELEX process was used to select aptamers capable of distinguishing the insecticide fenitrothion from non-specific targets with LOD of 14 nM [66].

Conventional analytical strategies for detecting pesticides are time-consuming processes that should be performed by trained personnel, which limits their use. Hence, the future of pesticide sensing lies in the development of devices enabling rapid and accurate on-site detection of pesticides or point-of-care analysis. Devices based on various portable detection technologies will enable effective on-site monitoring of pesticide residues in real samples [67]. Therefore, the search for reliable and promising molecular recognition elements remains a vital practical task. 

## 4. Materials and Methods

### 4.1. Reagents and Pesticides 

The study was performed using lyophilized enzymes: BChE from equine serum, 900 U/mg (Sigma-Aldrich, St. Louis, MO, U.S.A.); ADH from baker’s yeast, 300 U/mg (Sigma-Aldrich, St. Louis, MO, U.S.A.); ALP from bovine intestinal mucosa 10 DEA U/mg (Merck, Gillingham, Dorset, U.K.); trypsin from porcine pancreas, 1300 BAEE U/mg (Sigma-Aldrich, St. Louis, MO, U.S.A.); LDH from rabbit muscle, 600 U/mg (Sigma-Aldrich, St. Louis, MO, U.S.A.); Red from *Vibrio fischeri*, 0.15 U/mL (Institute of Biophysics, Siberian Branch of the Russian Academy of Sciences, Russia); and a mixture of high-purity enzymes: 0.15 U of Red from *Vibrio fischeri* and 0.5 mg of recombinant Luc *Photobacterium leiognathi* (Institute of Biophysics, Siberian Branch of the Russian Academy of Sciences, Russian Federation).

Bacterium *P. phosphoreum* 1889 was provided by the museum at the IBP SB RAS [68]. *P. phosphoreum* cells were grown for 24 h on solid medium for marine bacteria. The cells were suspended in the sodium-phosphate buffer pH 7.4.

The following reagents were used: NADH (Gerbu Biotechnik, Heidelberg, Germany), FMN (Serva, Heidelberg, Germany), Nα-Benzoyl-L-arginine ethyl ester (BAEE) (Sigma-Aldrich, St. Louis, MO, U.S.A.), pyruvate (Sigma-Aldrich, Tokyo, Japan), tetradecanal (Merck, Darmstadt, Germany), NAD^+^ (AppliChem, Darmstadt, Germany), S-BCh-I (Merck, Schaffhausen, Switzerland), 4-nitrophenyl phosphate disodium salt hexahydrate (Merck, Gillingham, Dorset, U.K.), 5.5′-Dithiobis(2-nitrobenzoic acid) (Sigma-Aldrich, Taufkirchen, Germany), MgCl_2_ (Sigma-Aldrich, Petaling Jaya, Malaysia), HCl (SigmaTek, Khimki, Russia), glycine NaOH buffer pH 9.6, sodium-phosphate buffer pH 7.4, Clark and Lubs buffer pH 7.6, potassium-phosphate buffer pH 6.8–8.0, and 95% ethanol. 

As analytes, we used 7 high-purity active ingredients of pesticides: fenvalerate, deltamethrin, cypermethrin, imidacloprid, malathion, diazinon, glyphosate (Sigma-Aldrich, St. Louis, MO, U.S.A.). Pesticide solutions were prepared using distilled water, acetonitrile 99.9% (PanReac AppliChem, Barcelona, Spain), or ethanol (95%) as solvents.

Bioluminescence was measured using a GloMax 20/20 luminometer (Promega Corporation, Madison, WI, U.S.A.) and a Lumat LB 9507 bioluminometer (Berthold Technologies, Bad Wildbad, Germany). To estimate the activities of single-enzyme assay systems and to investigate the spectral properties of active ingredients of pesticides, a Shimadzu UV-2600 spectrophotometer (Shimadzu Corporation, Kyoto, Japan) was used.

### 4.2. Effects of Active Ingredients of Pesticides on the Activities of Single-Enzyme Systems 

The activities of enzyme assay systems were determined by measuring the absorbance of reaction mixture solutions or bioluminescence intensity in the solutions of the analyzed pesticide active ingredients or in the control solution.

The activities of single-enzyme assay systems with Red, ADH, and LDH were determined by changes in the absorbance at 340 nm. 

The reaction mixture for the Red-catalyzed reaction consisted of 750 μL of the 0.05 M potassium-phosphate buffer pH 7.25, 6 mU of Red, 100 μL of the 0.4 mM NADH solution, 10 μL of the 0.5 mM FMN solution, and 5–100 μL of the analyte solution or the solvent (control). 

The activity of the ADH was estimated by using the following reaction mixture: 1500 μL of the potassium-phosphate buffer 0.05 M pH 7.85, 750 mU of ADH, 40 μL of the 2.4 mM NAD^+^ solution, and 25 μL of 95% ethanol. Distilled water and acetonitrile were used as control solutions. 

To analyze BChE activity Ellman’s method was used [69]. The absorbance of the solutions was measured at the 412 nm. We used the following reaction mixture: 800–850 μL of the 0.05 M potassium-phosphate buffer pH 8.0, 70 mU of BChE, 60 μL of 0.2 mM 5,5′-dithiobis (2-nitrobenzoic acid), 60 μL of 0.2 mM S-BCh-I, and 50–100 μL of the analyte solution or the solvent solution used as control. 

The reaction mixture for the LDH-catalyzed reaction comprised 850 μL of the 0.05 M potassium-phosphate buffer pH 8.0, 9 U of LDH, 40 μL of the 3.25 mM NADH solution, 30 μL of the 69 mM pyruvate solution, and 50 μL of the analyte solution or the solvent solution (control). 

The activity of the ALP was analyzed in the following reaction mixture: 990 μL of the glycine NaOH buffer pH 9.6 containing 0.5 mM MgCl_2_, 1.2 mU of the ALP solution, 8 μL of the 33 mM n-nitrophenyl phosphate solution, and 30–50 μL of the analyte solution. The absorbance of the solutions was measured at 405 nm.

Trypsin activity was analyzed in the reaction mixture containing 490 μL of the 0.1 M Clark and Lubs buffer pH 7.6, 11 mU of trypsin, 40 μL of 1 mM hydrochloric acid, and 460 μL of the 0.5 mM BAEE solution. The absorbances of the solutions were measured at 253 nm.

The effects of the active ingredients of pesticide and the solvents on the activity of the enzymes in the single-enzyme assay systems were determined as the relative activity according to the formula A = (A_t_/A_c_)∙100%, where A_c_ and A_t_ are the enzyme activity in the control and in the analyte solution, respectively.

### 4.3. The Effects of Active Ingredients of Pesticides on the Activity of Multi-Enzyme Systems 

The activities of the multi-enzyme assay systems were determined from the values of the luminescence intensity in the presence of the control or the analyte solutions.

To analyze the effect of pesticides on the luminescence intensity of the Red + Luc coupled enzyme assay system, we used the following reaction mixture: 290 μL of the 0.05 M potassium-phosphate buffer pH 6.8; 5 μL of the Red + Luc mixture, preliminarily diluted in 5 mL of the buffer solution; 50 μL of the 0.0025% tetradecanal solution; 50 μL of the 0.5 mM FMN solution; 100 μL of the 0.4 mM NADH solution; and 10–200 μL of the analyte solution or the solvent solution used as a control. Ratio of Red to Luc in the reaction mixture was 1.2:1; and their concentrations were about 15 and 13 nM, respectively.

The reaction mixture for analyzing the luminescence intensity of the LDH + Red + Luc three-enzyme assay system consisted of 5 μL of 0.5 mg/mL LDH; 300 μL of the 0.05 M potassium-phosphate buffer pH 7.1; 10 μL of the 15 mM lactate solution; 50 μL of the 0.0025% tetradecanal solution; 10 μL of the Red + Luc solution, preliminarily diluted in 5 mL of the buffer solution; 10 μL of the 0.5 mM FMN solution; 100 μL of the 0.5 mM NAD^+^ solution; and 10–50 μL of the analyte solution or the solvent solution (control). The proportions of LDH, Red, and Luc in the reaction mixture were 1.4:1.2:1, and their concentrations were about 36, 31, and 26 nM, respectively.

An analysis of the activity of the ADH + Red + Luc three-enzyme assay system was performed using the reaction mixture containing 350 μL of the 0.05 M potassium-phosphate buffer pH 6.9; 5 μL of 0.5 mg/mL ADH; 5 μL of the Red + Luc solution, preliminarily diluted in 5 mL of the buffer solution; 50 μL of the 0.0025% tetradecanal solution; 100 μL of the 0.4 mM NAD^+^ solution; 10 μL of the 0.5 mM FMN solution; 5 μL of 95% ethanol; and 10–100 μL of the solvent (control) or the analyte solution. The proportions of ADH, Red, and Luc in the reaction mixture were 10:1.2:1, and their concentrations were about 120, 14, and 12 nM, respectively.

The effect of the pesticide active ingredients and solvents on the multi-enzyme reactions was estimated from the residual luminescence intensity calculated according to the formula I = (I_t_/I_c_)∙100%, where I_c_ and I_t_ are the average values of luminescence intensity in the presence of the control or analyte solution, respectively.

The parameter IC_50_ was used to estimate the inhibitory effect of the active ingredients of pesticides on the enzyme activity and the bioluminescence intensity of a multi-enzyme assay system. It is the concentration of the active ingredient decreasing the enzyme activity by 50%.

### 4.4. The Effects of Active Ingredients of Pesticides on Bioluminescence of the Assay System Based on P. phosphoreum Luminous Bacterium 

The effects of active ingredients of pesticides on the luminous bacterium were determined by changes in the bacterial luminescence intensity in the presence of the analyte relative to the control value. The solvent was used as a control solution. The reaction mixture for analyzing the luminescence intensity of luminous bacterium in the presence of control solution contained 450 μL of bacterial suspension and 5 μL of ethanol or 50 μL of distilled water, used as solvents for active ingredients of pesticides. A cuvette with the reaction mixture was placed into a luminometer, and luminescence intensity (I_c_) was measured after 1 min. Then, another aliquot of bacterial suspension and 5 or 50 μL of a pesticide active ingredient solution were placed into a luminometer cuvette, and luminescence intensity (I_t_) was measured after 1 min. The residual luminescence intensity was determined according to the formula I = (I_t_/I_c_) ∙ 100%.

The parameter EC_50_ was used to estimate the inhibitory effect of the active ingredients of pesticides on the bioluminescence of *P. phosphoreum.* It is the concentration of the active ingredient decreasing the intensity of bioluminescence of the bacterium by 50%.

### 4.5. Fluorescence and Circular Dichroism Spectra Measurements 

Fluorescence emission of the enzymes in the absence and in the presence of commercial pesticide formulations was measured under excitation at 280 and 290 nm using a Cary Eclipse spectrofluorometer (Varian). The sample volume was 100 μL, and optical path length was 3 mm. At least three scans were averaged for each spectrum. Fluorescence spectra were corrected for PMT spectral sensitivity and background signal.

The absorption spectra in the 200–600 nm range were measured using a Cary 500 spectrophotometer (Varian) to estimate inner filter effect and, if necessary, correct it. 

The CD spectra were obtained using a Jasco-1500 spectropolarimeter (Jasco, Japan). The far-UV CD spectra were recorded for enzyme samples in a 0.1 mm path length cell in the range of 190–250 nm, with a step size of 2 nm. Spectra were baseline corrected. 

The measurements were performed at room temperature.

### 4.6. Statistical Analysis 

Each data point was the result of at least five measurements. The means and standard deviations were calculated for the maximum luminescence intensity (I_t_, I_c_) and enzyme activity (A_t_, A_c_). The results were statistically processed using the EXCEL software package (Microsoft, Redmond, Washington, U.S.A.). 

## 5. Conclusions

The current study investigated effects of commercial pesticide formulations and their active ingredients of various chemical classes (organophosphorus, pyrethroid, and neonicotinoid compounds) on different enzyme-based assay systems. Experiments demonstrated that AIfs often produce a substantially stronger inhibitory effect on enzyme activity than AIas. Similar results were obtained in experiments testing the effects of AIas and AIfs on bioluminescence intensity of the bacterium *P. phosphoreum*. These findings are consistent with the literature data and suggest a considerable contribution of formulants to the effect of pesticide formulations on the functions of enzyme assay systems and organisms. 

Circular dichroism and fluorescence spectra of the enzymes used for assays did not show any changes in the protein structure in the presence of commercial pesticide formulations during the assay procedure (i.e., without preliminary incubation). Only BChE fluorescence demonstrated sensitivity to the presence of the pesticide formulation, indicating an interaction between protein surface and pesticide formulation components, which requires further investigation. As the tertiary and secondary structures of enzymes were not disrupted, the inhibitory effect of pesticides on enzymes was caused by other mechanisms 

Thus, the present study shows that the toxic effect of pesticide formulations on enzymes is caused by a combination of molecular interactions and the contribution of inert compounds of commercial formulations to the total toxicological effect, which may lead to unpredictable consequences for non-target species. This supports the idea about the necessity of regulatory control in the use of formulants in agricultural practices.

## Figures and Tables

**Figure 1 ijms-24-02268-f001:**
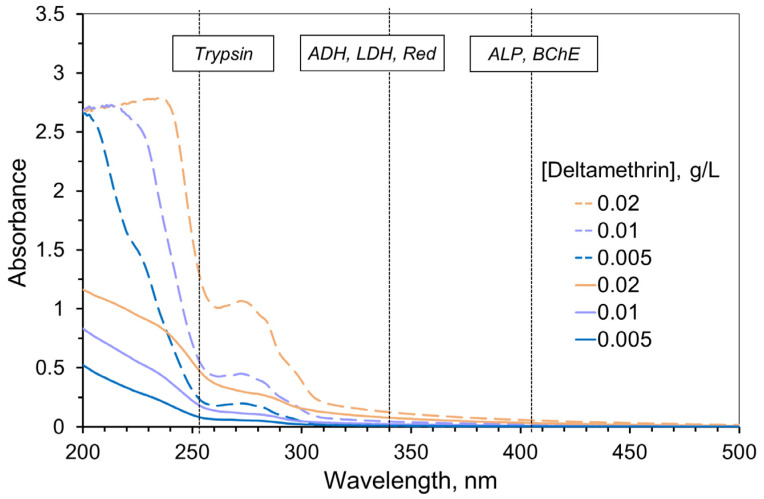
Absorption spectra of the deltamethrin solutions (solid lines) and commercial pesticide formulation Delcid with similar deltamethrin content (dashed lines) obtained with 1 cm optical path length. Acetonitrile was used as the solvent. Vertical dotted lines refer to the wavelengths used for measuring the activity of the enzymes. Bioluminescence intensity was measured in the 400–600 nm range. The absorption of the commercial pesticide formulation demonstrates higher optical density and indicates the need to take into account the inner filter effect when measuring enzyme activities, especially for the assay system with trypsin.

**Figure 2 ijms-24-02268-f002:**
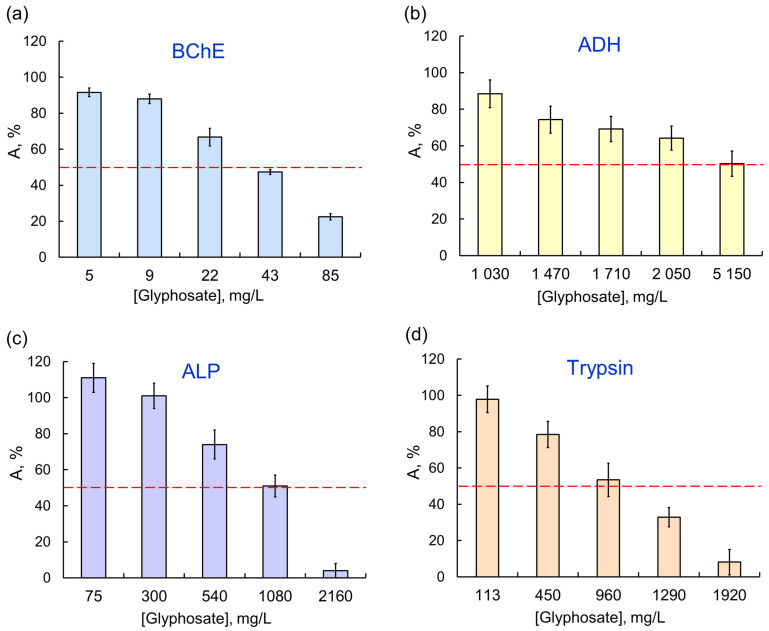
The effect of glyphosate as AIa on single-enzyme assay systems: (**a**) BChE; (**b**) ADH; (**c**) ALP; (**d**) trypsin. The assay systems differ in their sensitivity to the same AIa (glyphosate); the IC_50_ values for BCh, ADH, ALP, and trypsin are 0.035, 5.14, 1.08, and 0.96 μg/L, respectively.

**Figure 3 ijms-24-02268-f003:**
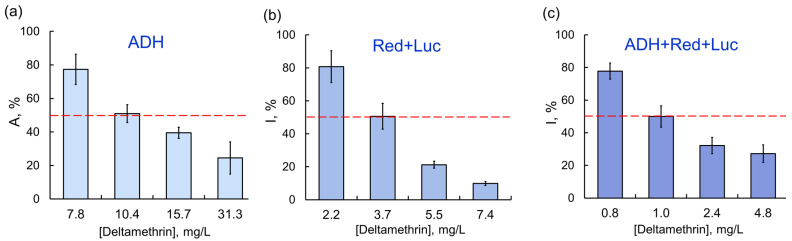
Relationship between residual activity of enzyme assay systems and concentration of deltamethrin as AIa: (**a**) ADH; (**b**) Red + Luc; (**c**) ADH + Red + Luc. The inhibitory effects of deltamethrin as AIa were enhanced by elongation of the chain of conjugated enzyme reactions.

**Figure 4 ijms-24-02268-f004:**
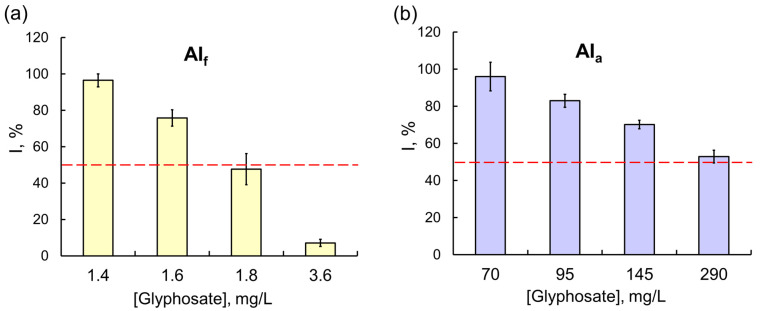
The effects of (**a**) the Tornado Extra commercial pesticide formulation (with glyphosate as the active ingredient) and (**b**) glyphosate as AIa on the Red + Luc coupled enzyme system. Glyphosate as AIf had a much stronger inhibitory effect on the Red + Luc assay system compared with glyphosate as AIa; the values of IC_50_ for glyphosate as AIf and AIa were 1.8 and 288 mg/L, respectively.

**Figure 5 ijms-24-02268-f005:**
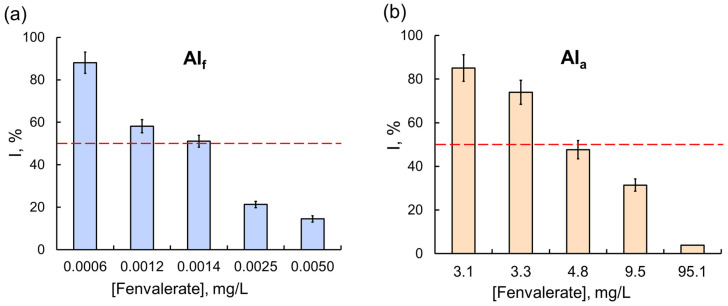
The effects of (**a**) the Sempay commercial pesticide formulation (with fenvalerate as the active ingredient) and (**b**) fenvalerate as AIa on the Red + Luc coupled enzyme system. The same as for glyphosate, the inhibitory effect of fenvalerate as AIf on the Red + Luc system was stronger than the effect of fenvalerate as AIa; the values of IC_50_ for fenvalerate as AIf and AIa were 0.0014 and 4.8 mg/L, respectively.

**Figure 6 ijms-24-02268-f006:**
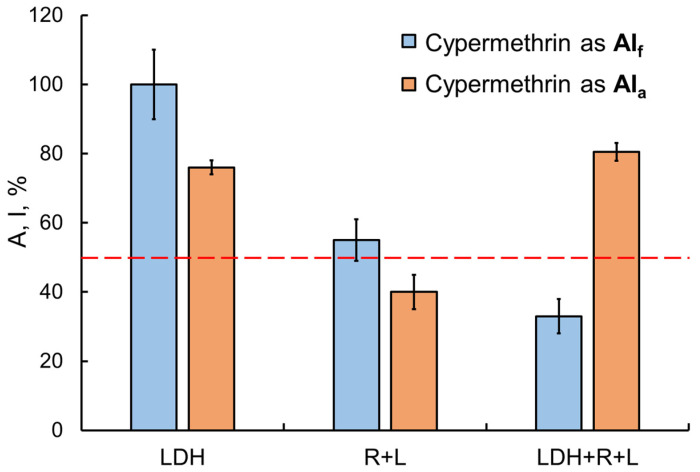
Comparison of the effects of the Briz commercial pesticide formulation (with cypermethrin as the active ingredient) and cypermethrin as AIa on enzyme assay systems with different lengths of the enzyme conjugation chain. Cypermethrin concentration was 2 mg/L. The inhibitory effects of cypermethrin as AIf were enhanced by elongation of the chain of conjugated enzyme reactions. No such dependence, though, was found for cypermethrin as AIa.

**Figure 7 ijms-24-02268-f007:**
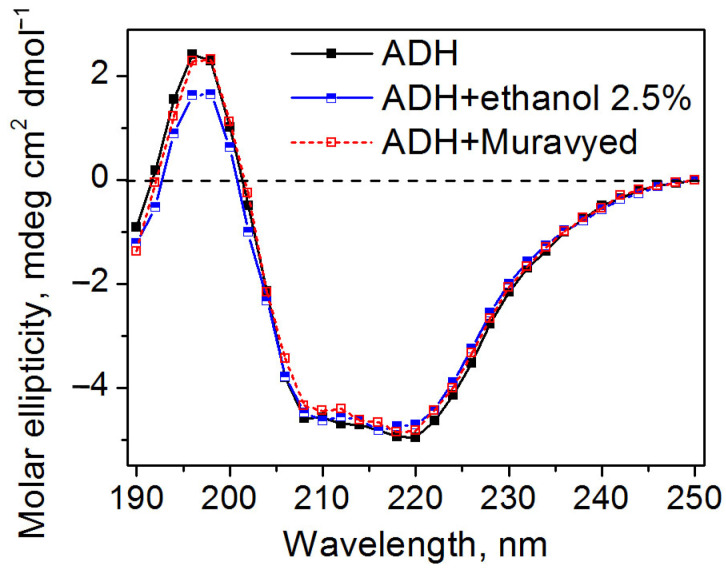
Circular dichroism spectra of ADH without additives (black) and in the presence of ethanol (blue) and the Muravyed commercial pesticide formulation (red). ADH concentration was 0.1 mg/mL, diazinon concentration in Muravyed was 0.02 mg/mL. Neither ethanol nor Myravyed alters the secondary structure of ADH.

**Figure 8 ijms-24-02268-f008:**
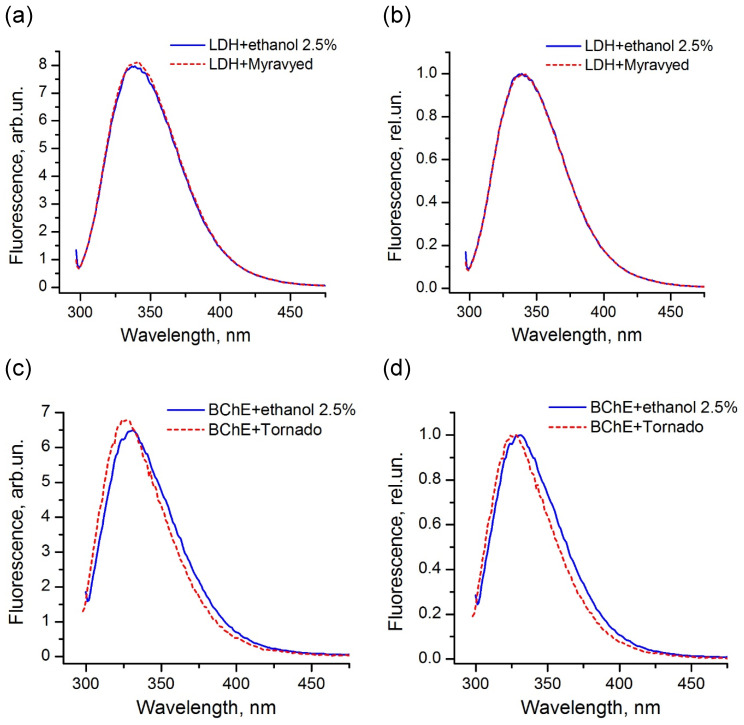
Fluorescence spectra of LDH (**a**,**b**) and BChE (**c**–**f**) under 290 nm excitation in the presence (dashed lines) and in the absence (solid lines) of commercial pesticide formulations Muravyed (**a**,**b**), Tornado Extra (**c**,**d**), and Biotlin (**e**,**f**). Normalized (**b**,**d**,**f**) and not normalized (**a**,**c**,**e**) spectra are shown. Protein concentrations were 0.4 mg/mL of LDH, 0.25 mg/mL of BChE. Concentrations of diazinon in Muravyed, glyphosate in Tornado Extra, and imidacloprid in Biotlin were 1.25 mg/L, 2.4 mg/mL, and 0.02 mg/mL, respectively. For LDH, no effect of Muravyed on protein fluorescence was observed. The components of Tornado Extra and Biotlin formulations cause the blue shift of the BChE fluorescence.

**Table 1 ijms-24-02268-t001:** IC_50_ (mg/L) and EC_50_ (mg/L) values determined from the effects of AIas and AIfs on the activities of enzyme-based assay systems and *P. phosphoreum* bioluminescence.

Assay System		Fenvalerate		Deltamethrin		Cypermethrin		Imidacloprid				Malathion		Diazinon		Glyphosate		
		AIa	AIf Sempay	AIa	AIf Delcid	AIa	AIf Briz	AIa	AIf			AIa	AIf Aliot	AIa	AIf Muravyed	AIa	AIf Liquidator	AIf Tornado Extra
									Biotlin	Corado	Confidor Extra							
Single-Enzyme Assay Systems	Trypsin	x	*	x	–	–	*	–	*	*	–	*	*	*	*	962	5400	2400
	ALP	x	*	x	–	–	–	–	*	*	*	*	*	*	*	1080	600	220
	BChE	120	-	100	0.76	–	30,930	–	200	–	80,000	600	4	–	20	35	1000	2.4
	LDH	3	0.2	30	6.2	25	150	–	–	180	1	350	30	–	0.05	–	6000	52
	ADH	–	*	10.4	16.7	0.2	100	–	0.17	0.08	49.9	*	*	14.5	0.2	5140	1.5	2.1
	Red	–	*	–	146	–	300	–	0.09	–	14.9	*	*	*	*	–	9.0	5.0
Multi-Enzyme Assay Systems	Red + Luc	4.8	0.0014	3.7	39.5	1.8	5	–	0.003	0.07	34.4	*	0.1	2234	0.009	288	1.11	1.8
	ADH + Red + Luc	1.6	0.0006	1.0	12.7	0.2	3	–	0.006	0.04	47.8	*	0.05	11	0.01	3200	1.4	2.0
	LDH + Red + Luc	31.7	0.0007	7.7	11.5	6.5	1	–	0.01	0.04	1.9	*	0.014	3351	0.005	935	1.1	3.3
*P. phosphoreum*		–	*	–	*	–	*	–	2000	500	110	–	*	–	*	–	400	400
MRL RUS mg/kg [27]		0.02–0.1		0.01–0.3		0.01–2.0		0.1–1.0				0.05–1.0		0.1–0.5		0.1–5.0		

«*» The parameter could not be determined because of physico-chemical properties of the AIa and AIf or interaction of the AIa and AIf with the reaction mixture components. «–» No inhibitory effect of the AIa and AIf was detected in the tested concentration range. «x» A stimulating effect of the active ingredient on parameters of assay systems was observed in the tested concentration range.

## Data Availability

The data presented in this study are available in article.

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
