# Peer review of "Enzyme Inhibition-Based Assay to Estimate the Contribution of Formulants to the Effect of Commercial Pesticide Formulations"

_ijms, 2023, doi:10.3390/ijms24032268_

Round 1
Reviewer 1 Report
The safety of “inert” ingredients is often ignored by people. In this paper, the contribution of formulants to toxicological properties of the pesticide formulations has been studied by analyzing effects of 7 active ingredients of pesticides and 10 commercial formulations based on them on the function of a wide range of enzyme assay systems differing in complexity (single-, coupled, and three-enzyme assay systems).
1. In Figure 1, in order to facilitate reading, please turn up the words and numbers, and show more numbers in the ordinate.
2. In Figures 2, 3, 4 and 5, please align the marks a, b... on the upper left corner of each figure.
3. In particular, please show the amount and proportion of enzymes in the two enzyme system test and the three enzyme system test in the Figure and Table.
4. The dosage of pesticide concentration does not show a gradient ratio relationship. How did the author consider it when testing? For example, in Figure 3 (b), the concentration of deltamethrin is 2.2,3.7,5.5,7.4 mg/L.
Author Response
Dear Reviewer,
Thank you for your comments and suggestions. Here are our answers and corrections.
Remark 1: In Figure 1, in order to facilitate reading, please turn up the words and numbers, and show more numbers in the ordinate.
Answer 1: We have moved the words and numbers higher and added more numbers to the ordinate axis.
Remark 2: In Figures 2, 3, 4 and 5, please align the marks a, b... on the upper left corner of each figure.
Answer 2: We have aligned the marks a, b, etc. in Figures 2-5 as suggested.
Remark 3: In particular, please show the amount and proportion of enzymes in the two enzyme system test and the three enzyme system test in the Figure and Table.
Answer 3: We have added the requested information in part 4.3. The Effects of Active Ingredients of Pesticides on the Activity of Multi-Enzyme Systems.
In particular:
Lines 649-650: Ratio of Red to Luc in the reaction mixture was 1.2 : 1; and their concentrations were about 15 and 13 nM, respectively.
Lines 656-658: The proportions of LDH, Red, and Luc in the reaction mixture were 1.4 : 1.2 : 1; and their concentrations were about 36, 31, and 26 nM, respectively.
Lines 664-666: The proportions of ADH, Red, and Luc in the reaction mixture were 10 : 1.2 : 1; and their concentrations were about 120, 14, and 12 nM, respectively.
Remark 4: The dosage of pesticide concentration does not show a gradient ratio relationship. How did the author consider it when testing? For example, in Figure 3 (b), the concentration of deltamethrin is 2.2,3.7,5.5,7.4 mg/L.
Answer 4: We did not have a purpose to show a gradient ratio relationship. The main aim of the experiment presented on figure 3 was to determine values of IC50 for every of enzyme assay system. To do this, we mixed the initial solution of the pesticide and the solvent solution in different proportions.
Reviewer 2 Report
The article titled “Enzyme inhibition-based assay to estimate the contribution of formulants to the effect of commercial pesticide formulations” submitted by Esimbekova et al., is interesting. The authors used multiple enzyme-based comparative studies to examine the different active ingredients and commercial pesticide formulations with a focus on the molecular level. They also compared the results to experiments based on bacterial bioluminescence. These results are interesting, and the manuscript is well-written and well-structured. I believe that the objectives of the study are interesting and fit well with the scope of the journal and that the analysis was carried out with an adequate description of the methods used.
Author Response
Dear reviewer,
Thank you for your appreciation of our research.
Reviewer 3 Report
1. The abbreviations: for AIa and AIf, need to be better written.
2. Authors could briefly introduce novel DNA-aptamerics biosensors developed for pesticides, as in https://doi.org/10.1016/j.teac.2022.e00184; and discuss more studies by these authors on fipronil, malathion, diazinon, and fenitrothion. Compare these aptameric sensors with CD spectral analysis of ADH and pesticides.
3. For Figure captions, please expand the captions to describe the content of each figure. The captions should convey key message any have meaningful information. Difficult to understand the figures.
4. Describe significance of CD and fluorescence results; compare them with previous studies in discussion section.
5. Compare the performance (LOD/Kd) of sensors vs other target specific biosensors.
6. Provide perspective and future directions for pesticide sensing.
7. Although, the concept is relatively old; the article is interesting and have significance for pesticidal analysis, however, it must be carefully revised as per above suggestions.
Author Response
Dear Reviewer,
Thank you for your comments and suggestions. Here are our answers and corrections.
Remark 1: The abbreviations: for AIa and AIf, need to be better written.
Answer 1: We have more clearly marked the indices of these abbreviations throughout the text.
Remark 2: Authors could briefly introduce novel DNA-aptamerics biosensors developed for pesticides, as in https://doi.org/10.1016/j.teac.2022.e00184; and discuss more studies by these authors on fipronil, malathion, diazinon, and fenitrothion. Compare these aptameric sensors with CD spectral analysis of ADH and pesticides.
Answer 2: Indeed, recently many DNA-aptameric biosensors have been constructed to detect various substances, including pesticides, with high selectivity and affinity. In case of DNA-aptameric biosensors, circular dichroism spectroscopy is applied to determine the binding affinity of DNA fragments against target molecules. In our study, the aim of CD spectra was to revel the damage of the protein secondary structure due to pesticides presence, if any. No specificity of the enzymes used to the additives were expected. So, our strategy is slightly different, but common for protein study. CD spectra of ADH presented in Fig. 7 demonstrate no effect of pesticides on enzyme structure, which was observed for all studied enzymes.
We have added to the discussion section the fragments about aptamers:
Lines 555-564: Aptamers (short nucleotide sequences of single-stranded ribonucleic or deoxyribonucleic acids) are used as the basis for developing specific, measurable, accurate, robust, and time-saving (SMART) biosensors – aptasensors. They demonstrate high selectivity in binding with targets and remain functionally active during long-term storage, even at room temperature [63]. Higher stability, longer lifetime, and lower cost are advantages of aptamers over enzymes and antibodies [62]. Aptasensors exhibited high sensitivity to pesticides such as fipronil [64], diazinon [65], chlorpyrifos [66], and acetamiprid [67] in real samples (fruits, vegetables, wastewater). The SELEX process was used to select aptamers capable of distinguishing insecticide fenitrothion from non-specific targets with LOD of 14 nM [68].
Lines 509-514: CD spectra provide different information depending on the biological recognition element. For example, for aptamer-based sensors, circular dichroism spectroscopy is used to estimate the binding affinity of nucleic acid fragments against a certain pesticide [56], whereas in enzyme inhibition-based assays, it is applied to detect a general change in the protein secondary structure, without any specificity [50,57].
Remark 3: For Figure captions, please expand the captions to describe the content of each figure. The captions should convey key message any have meaningful information. Difficult to understand the figures.
Answer 3: We have describe the content of each figure. Now all of them have meaningful information.
Remark 4: Describe significance of CD and fluorescence results; compare them with previous studies in discussion section.
Answer 4: We have added a more detailed description of the obtained CD and fluorescence spectra with 6 new references:
Lines 313-317: The examples of CD spectra of ADH in the presence of the Muravyed commercial pesticide formulation or ethanol, as well as without any additives, are shown in Figure 7. All CD spectra of the ADH were found to have double minima at 208 and 220 nm, which is consistent with previously published data [30] and reflects the well-known α-helical structure of this protein.
Lines 337-344: The peak intensity of the protein fluorescence spectrum was observed at about 341 nm in the presence of both ethanol (control) and pesticide formulation. However, the fluorescence of BChE changed in the presence of both tested formulations – Tornado Extra and Biotlin. After addition of the Tornado Extra formulation, spectral maximum of BChE fluorescence was blue-shifted from 331 to 327 nm without intensity change as compared with control sample (Figure 8, c and d). In the experiment with Biotlin, essential quenching of BChE fluorescence was observed (Figure 8, e), with a slight blue shift of the spectral maximum to 329 nm (Figure 8, f).
Lines 506-530:
CD spectra of ADH, LDH, BChE, ALP, and trypsin were not altered in the presence of the pesticide formulations, although some studies using this technique revealed structural changes of proteins, e.g. of human serum albumin by fungicide carbendazim [54], of pepsin by pyrethroid insecticides [55], etc. CD spectra provide different information depending on the biological recognition element. For example, for aptamer-based sensors, circular dichroism spectroscopy is used to estimate the binding affinity of nucleic acid fragments against a certain pesticide [56], whereas in enzyme inhibition-based assays, it is applied to detect a general change in the protein secondary structure, without any specificity [50,57].
The change of the intrinsic protein fluorescence in the presence of different xenobiotics is extensively used to study the action mechanisms of the toxic substances. The decrease in fluorescence intensity under variation of the temperature and additive concentration can be used as the basis for estimating the affinity and thermodynamic characteristics of protein-xenobiotics interaction (see [54,55,57] as examples). However, to the best of our knowledge, the direct interactions between enzymes and pesticides used in our work have never been studied.
Three reasons could be proposed to explain the absence of denaturing effect of the studied pesticide formulations on the enzyme structure: (i) low concentration of the additives; (ii) a stabilizing effect of other than pesticide components of the formulations; and (iii) too short time of incubation of the proteins with additives. Since this part of our study was aimed at elucidating the mechanism of the observed inhibitory action of the pesticide formulations, the experimental conditions were the same as those under which the activity of the studied enzymes was measured. A wider variation of the experimental conditions could result in pronounced disruption of protein structure by pesticide formulations, but this would be the subject of further detailed research.
Remark 5: Compare the performance (LOD/Kd) of sensors vs other target specific biosensors.
Answer 5: In this research article, we only assessed the potential of the enzyme assay systems for pesticide detection. We did not set out to develop a biosensor. For this reason, we cannot compare sensor characteristics as suggested. To avoid enlarging the text of the manuscript, we have given the LOD values for several modern biosensors used for detecting pesticides on the lines 540-549.
Remark 6: Provide perspective and future directions for pesticide sensing.
Answer 6: We have added information about other biosensors and discussed their perspective for pesticide sensing.
Lines 536-571:
The knowledge of the mechanisms of pesticide molecular action forms the basis for the methods of monitoring pesticide residues using biosensor processes, which, in addition to enzymes, employ such molecular recognition elements as antibodies, nucleic acids, aptamers, etc. [58].
The enzymes that are commonly used to detect pesticides include hydrolases AChE, BChE, alkaline phosphatase, lipase, as well as oxidoreductases horseradish peroxidase, tyrosinase, and laccase. Electrochemical biosensors based on AChE and horseradish peroxidase were effectively used to detect OPs: detection limits were 0.16 ng/mL of malathion and 0.025 mg/L of glyphosate, respectively [59,60]. In the current study, among the single enzyme systems, the enzyme assay system with ADH exhibited the highest sensitivity to another OP pesticide – diazinon. The values of IC50 for diazinon as AIa and AIf were 14.5 and 0.2 mg/L, respectively. Enzyme biosensors based on multi-enzyme systems show considerable promise as well [61]: they exhibit high sensitivity to toxic substances, as confirmed by results of the present study.
The principal advantages of immunosensors over the enzyme-based biosensors are the higher stability of antibodies/antigens used as recognition elements and greater selectivity and specificity. Modifications with different (nano)materials and the use of enzymatic tags make it possible to produce diverse immunosensors, which are capable of detecting pesticides in real food samples [62].
Aptamers (short nucleotide sequences of single-stranded ribonucleic or deoxyribonucleic acids) are used as the basis for developing specific, measurable, accurate, robust, and time-saving (SMART) biosensors – aptasensors. They demonstrate high selectivity in binding with targets and remain functionally active during long-term storage, even at room temperature [63]. Higher stability, longer lifetime, and lower cost are advantages of aptamers over enzymes and antibodies [62]. Aptasensors exhibited high sensitivity to pesticides such as fipronil [64], diazinon [65], chlorpyrifos [66], and acetamiprid [67] in real samples (fruits, vegetables, wastewater). The SELEX process was used to select aptamers capable of distinguishing insecticide fenitrothion from non-specific targets with LOD of 14 nM [68].
Conventional analytical strategies for detecting pesticides are time-consuming processes that should be performed by the trained personnel, which limits their use. Hence, the future of pesticide sensing lies in the development of devices enabling rapid and accurate on-site detection of pesticides or point-of-care analysis. Devices based on various portable detection technologies will enable effective on-site monitoring of pesticide residues in real samples [69]. Therefore, the search for reliable and promising molecular recognition elements remains a vital practical task.
Remark 7: Although, the concept is relatively old; the article is interesting and have significance for pesticidal analysis, however, it must be carefully revised as per above suggestions.
Answer 7: We have tried to revise the manuscript according to the suggestions.
Round 2
Reviewer 3 Report
Authors have addressed the queries well.